# From Bitter to Better Lessons in AI: Embracing Human Expertise as Data

## Abstract

Artificial intelligence (AI) and machine learning (ML) have long treated data as clean numeric features and labels, with progress driven by ever-larger models and datasets, a view that is crystallized in Sutton's "Bitter Lesson". In this paper, we contend that human expertise, often encoded in natural language, mathematical formalisms, and software, should itself be regarded as a vital form of data. First, we survey physics-informed ML, geometric deep learning, and safe reinforcement learning to show how embedding expert knowledge narrows hypothesis spaces, reduces sample and computational complexity, and improves out-of-distribution generalization. Next, we trace the expanding scope of data in ML, demonstrating how integrating text, images, actions, and other data modalities can transform previously transductive learners into increasingly inductive ones. We then highlight large language models (LLMs) as the nexus of these trends, illustrating how reinforcement learning with human feedback and in-context learning let LLMs integrate human expertise as data for general-purpose computation. To measure current practice, we analyze 1,000 NeurIPS papers between 2020–2024, finding that explicit domain-expert integration remains low with 12–18%, while LLM-based methods for expert incorporation are surging from 1% in 2022 to 8% in 2024. We revisit the Bitter Lesson amid slowing Moore's Law and real-world, non-i.i.d. data challenges, survey alternative perspectives, and propose new directions for dataset documentation, model design, and curated knowledge repositories. By recognizing human domain expertise and insights about tasks as first-class data, we envision a foundation for the development of more efficient and powerful AI.

## 1 Introduction

When we think about data in the context of artificial intelligence (AI) and its subfield of machine learning (ML), we often perceive these as clean numerical representations of features and labels. We often utilize such data in substantial volumes to effectively train prediction models that can solve a wide range of important real-world problems. Even when tackling complex scenarios involving audio, video, or natural language, we can represent our data through structured numeric frameworks, and increasingly also process them all at the same time using a combination of multiple specialized neural network architectures. To facilitate a stable training process, we typically normalize or standardize our data into a consistent numeric scale while maintaining essential information.

In this article, we argue that established practices like these often prevent us from recognizing alternative forms of information as crucial to effective problem-solving with ML. We argue that our growing ability to handle diverse data types has paved the way for a new era in AI. In this era, we can do more than merely leverage increased computational power and larger datasets; we can also incorporate contextual information and insights about a task as data. Often, such information consists

of human expertise residing in natural, mathematical, or programmed languages that provide valuable structure for solving more complex problems with ML than has previously been possible.

**We emphasize that integrating problem-specific insights into problem-solving with ML can significantly reduce the demand for computation and data. In this context, we highlight the emergence of large language models (LLMs) as powerful tools capable of integrating a diverse range of human expertise and context information about a task as data.** We ultimately conclude that these advancements may require new data collection strategies and model architecture designs, in order to unlock the full potential for a new, more sophisticated era of problem-solving in AI.

Our arguments become pivotal in light of Richard Sutton's influential perspective, *The Bitter Lesson*, published in 2019 [1]. Sutton contends that in the history of AI research, general computational methods have consistently outperformed approaches heavily reliant on human domain expertise. This trend is largely attributed to the exponentially decreasing cost of computation as described by Moore's Law. Sutton illustrates his point with examples from computer chess, Go, speech recognition, and computer vision, highlighting that strategies emphasizing deep *search* and *learning* eventually surpassed methods that incorporated human knowledge.

While we agree with Sutton's argument about major breakthroughs in AI predominantly resulting from scaling computational resources rather than mimicking human cognitive processes, we argue that his dismissal of human domain expertise as insignificant for AI-driven problem-solving is increasingly outdated, especially under a clear slow-down and approaching end to Moore's Law for digital computers. We base our counterargument on both the emerging capability and growing reliance of contemporary LLMs on expert domain knowledge as integral to general-purpose computation, alongside the rapid progress in ML models' abilities to digest and generate multiple data types simultaneously. These recent developments represent a paradigm shift unforeseen by Sutton and other prominent AI researchers, and surface an intrinsic limitation in their original perspectives.

In the following sections, we begin by emphasizing two key trends in AI and ML research. First, infusing ML models with problem-specific knowledge can reduce the need for computation and data, while often improving out-of-distribution generalization. Second, as the scope of data in ML grows, models gain the capacity to make more sophisticated predictions. We then show how LLMs unite these two trends by utilizing human domain expertise as data for general-purpose computation. Next, we offer a quantitative survey of NeurIPS papers from 2020-2024, reconsider Sutton's *Bitter Lesson* in an era of slowing Moore's Law, and survey alternative perspectives on these developments. We conclude our position with important implications that these developments have for existing data collection strategies and model architecture designs.

## 2 Human expertise can reduce the need for computation and data, and improve out-of-distribution generalization

A recurring theme in ML research is that integrating expert domain knowledge significantly reduces the number of data points required for solving a given problem. Technically speaking, incorporating such knowledge reduces the *sample complexity* or equivalently, increases *sample efficiency*, meaning that fewer data points are needed to achieve a particular level of model performance. When we apply these principles to deep *learning* or *search*-based methods, this typically results in a decrease of computational demand and thereby enhances our *computational efficiency*.

Integrating problem-specific knowledge into solution methods often also improves a model's ability to generalize beyond the training data distribution. Technically speaking, this enhancement reduces *epistemic uncertainty*, the uncertainty stemming from limited knowledge about novel conditions in which we want to solve a problem, and strengthens *out-of-distribution* generalization capabilities.

In the following, we illustrate these principles through three example areas of modern ML research: *physics-informed ML, geometric deep learning (geometric DL)*, and *safe reinforcement learning (safe RL)*.

**Physics-informed ML** refers to the integration of physical properties of a prediction task directly into the design of ML algorithms [2]. Current approaches utilize three distinct strategies: *inductive bias*, *observational bias*, and *learning bias*. Methods employing *inductive bias* embed physical knowledge directly into the structure of ML models, for instance, through specialized neural network architectures and *implicit layers* [3]. By encoding such physical knowledge, physics-informed ML

effectively narrows the hypothesis space that models must explore to fit patterns in the training data, leading to faster convergence and enhanced learning efficiency with fewer training samples.

An illustrative example of this is the application of physics-informed RL to optimize the freeform design of nanophotonic devices, a combinatorial optimization problem encompassing approximately $10^{74}$ candidate solutions [4]. Given a fixed budget for computation and data, Park et al. [4] show that physics-informed RL can reduce sample complexity by roughly 50% to match the performance of the best conventional RL approach. Furthermore, under an equivalent data usage, physics-informed RL outperforms conventional RL models by 7.8% in terms of *deflection efficiency* of the predicted nanophotonic designs.

**Geometric DL** is an emerging field of ML research that studies geometric properties of ML tasks and datasets [5]. Given a set of group symmetry transformations towards which we want data representations and predictions of a task to be either invariant or equivariant, geometric DL explores how we can incorporate these symmetries into neural network architectures. By exploiting domain-specific knowledge about prevalent symmetries in a task, we are able to shrink the problem volume space by choosing the right model architecture [6].

A concrete example that illustrates this well are Group Equivariant Convolutional Neural Networks (G-CNNs), which consider rotation and reflection symmetries for processing images. G-CNNs effectively learn robust representations with fewer samples compared to standard Convolutional Neural Networks (CNNs), which inherently handle only translation symmetries [7]. In an image classification task, for instance, G-CNNs enable robust detection of objects irrespective of their orientation, eliminating the need for extensive data augmentation with rotated images, which is a common practice when using traditional CNNs. Similarly, graph neural networks (GNNs) [8] and attention mechanisms in transformer architectures [9] capture permutation symmetries, and can thereby reduce the sample complexity of different sets of tasks where such symmetries are prevalent.

Analogous to physics-informed ML, we often observe that incorporating symmetry into ML models also enables better generalizations outside the distribution of data we are able to collect and use for training. Although a theoretical relationship is yet to be established, geometric DL methods appear to be closely related to physics-informed ML methods.

**Safe RL** is another pivotal yet distinct field of modern ML where expert domain knowledge can improve sample efficiency. Safe RL is a subfield of RL that deals with algorithms for decision-making and learning in complex and unpredictable environments while ensuring safety and minimizing the risk of harmful actions during the learning process, known as *exploration*. Methods typically incorporate domain-specific knowledge about ethical guidelines, physical limitations, and other criteria that constitute 'safe' behavior in a given task. These rules can often be incorporated into simulation models that provide a safe environment for the RL agent to explore and learn with less real-world data and fewer safety violations for performing a task.

A concrete application that illustrates this well is the training of RL policies for autonomous systems in simulated environments before real-world deployment, where safety violations are reduced by up to 77% when agents operate in novel environments [10]. In this context, the interactions of an RL agent with a previously unseen environment represent the generalization performance of the agent's actions to data that lies outside the distribution that the agent was able to encounter during training.

# 3 The scope of data in ML is growing, and increases the inductive power of models

The scope of what constitutes data in ML has been consistently expanding over time. As a result, ML models are evolving as not just being able to effectively process increasingly large amounts of data, but also effectively process increasingly diverse types and combinations of information as data, i.e., the *modality of data* associated with a given task.

By integrating diverse data sources into a prediction task, we can also assemble richer constellations of inputs and outputs for a markedly greater predictive power. In practice, this integration often converts *transductive* models, which may only predict a target when that same quantity has been observed, into *inductive* models, which can generalize in the absence of observations of that quantity.

A concrete application that illustrates this well is electric load profile forecasting. Traditionally, ML methods for this task are *auto-regressive*, meaning that they forecast future values of a load time series solely from its historical data using a model that is able to process sequential data only. As a result, these models are transductive and limited to making predictions for load that has historic measurements available as features. By contrast, recent multi-modal approaches integrate diverse data sources, like meteorological observations and satellite imagery, using specialized neural network architectures for each modality. These models take only remote-sensing inputs as features and metered load profiles as labels [11]. As a result, they can form inductive models that generalize to load profiles at unmetered locations using only readily available, remotely sensed meteorological and satellite data, without requiring ground truth load measurements to be available as features.

A more general example of this trend is the rise of multi-modal transformer architectures, which convert diverse inputs, such as text, images, actions, audio, and more into a unified sequence of tokens processed by a single model. The Gato model by Reed et al. [12], for instance, is trained on over 600 distinct tasks spanning dialogue, vision and robotic control, yet executes any of them through the same transformer backbone. By aligning different modalities in a shared representation space, these models demonstrate how broadening the notion of data empowers inductive learning, where a single model can generalize across tasks and domains far beyond the scope of single-modality models.

# 4 LLMs can utilize human expertise as data for general-purpose computation

The increasing capability of ML models to simultaneously process multiple data modalities as both inputs and outputs is also pivotal to incorporating human expertise as data for general-purpose computation. Human expert knowledge often resides in mathematical formulas, software, or natural language descriptions of important relationships [13]. This is information that we traditionally do not recognize as data for problem-solving when focusing on more conventional numeric values associated with training our ML models. In particular, we observe how LLMs have emerged as a powerful tool for integrating various types of human domain knowledge [14] and thereby expanded our scope of data for problem-solving with ML in an unprecedented way. Today, there are two approaches in which we observe this integration to unfold, namely through training and context data.

**Human expertise as training data.** The first approach involves *Reinforcement Learning with Human Feedback* (RLHF) [15]. Advanced language models such as ChatGPT owe much of their success to this technique. The training process of these models begins with a transformer architecture trained in an unsupervised manner on a large corpus of text to produce a *Generative Pretrained Transformer* (GPT). This model is then fine-tuned using supervised learning on a curated dataset of labeled question-answer pairs to form an initial dialogue agent. Subsequently, human feedback is introduced, typically by ranking or evaluating multiple model responses to the same prompt, to guide the model's behavior. Finally, the model's policy, its strategy for generating outputs, is optimized using RL to align with human preferences.

This pipeline highlights the essential role of human expertise as data at every stage, starting with unsupervised pre-training on carefully curated, high-quality text corpora, through supervised fine-tuning on labeled question-answer pairs, and culminating in RL guided by expert feedback.

**Human expertise as context data.** The second approach involves *in-context learning*. This is an approach in which an LLM learns to perform a task based on examples of solutions provided as input data in the prompt. A benefit of this method is that LLMs do not need explicit retraining or fine-tuning for that specific task and can usually learn from a few examples included in the prompt.

An example that illustrates this well is the use of in-context learning for mathematical optimization [16]. By providing several numeric samples that specify the task, along with a previously generated set of solutions and a corresponding score that captures how well the objective is satisfied, LLMs can leverage the optimization trajectory for recognizing patterns that guide the generation of new solutions along that trajectory. Experiments show that concurrent LLMs can solve a problem with fractions of only 3.5% and 20% of the available numeric datasets. This is notably also consistent with our observation that expert domain knowledge can reduce the need for computation and data.

We can expect that in-context learning reduces sample complexity either when tapping into interrelated patterns and domain knowledge that is included in an LLM's training data, or when these are explicitly provided as context data in a prompt.

## 5 A quantitative analysis of NeurIPS 2020 to 2024 studies

We want to quantifying how frequently practitioners weave problem-specific domain expertise into their ML models for more efficient solutions, and how often they employ LLMs to do so. A quantitative answer to these questions provides us with further insight about the trends we observe across the broader ML research landscape. We therefore conducted a representative survey of 1,000 NeurIPS papers published from 2020 through 2024, randomly sampling 200 papers per year. This interval spans both the advancement of domain expert integration methods and the adoption of LLMs, marked by the launch of GPT-3 in mid-2020 and ChatGPT in late 2022. For each paper, we provide its full text in PDF format to OpenAI's ChatGPT o4-mini-high model, and prompt this model as follows:

```
Review the attached research paper and address the following two questions:

1. Does the study incorporate domain-specific expert knowledge about a
problem into machine learning models to reduce the demand for computation
and/or data?

2. Are large language models (LLMs) used for incorporating domain-specific
expert knowledge as training or context data into problem solving?

Provide a "yes" or "no" answer and a one sentence explanation.
```

The requested one sentence explanation for each answer allows us to more easily examine the accuracy of each answer. We manually validate the correctness of all responses by checking the consistency of binary responses and provided explanations with the content of each paper. Additionally, we have iteratively improved our prompt in an LLM-driven optimization procedure, and ultimately selected the above prompt for its low false-positive and false-negative rates. To ensure that each evaluation is uncontaminated by prior context, we disable both memory and cross-chat references. An example of the model's dual-question response is:

```
Yes. The authors leverage neuroscience-inspired domain knowledge-
specifically the all-or-none firing characteristics and inter-/
intra-neuron temporal dependencies of spiking neurons-to design
the TSSL-BP algorithm, reducing the number of time steps (and thus
computation and data) needed to train deep SNNs effectively.

No. The work focuses solely on training deep spiking neural networks
via backpropagation and does not employ any large language models for
incorporating expert domain knowledge.
```

Table 1 contains the numeric results of our analysis, representing the proportion of papers falling into each response category for each year. We can observe that studies which incorporate human domain expertise into specialized ML models ("Yes–No" and "Yes–Yes") consistently account for only 12–18.5% of the sample, with no clear upward or downward trend over time. In contrast, the proportion of papers leveraging LLMs to encode expert knowledge ("No–Yes" and "Yes–Yes") increases noticeably following GPT-3's debut in mid-2020 and ChatGPT's release in late 2022, rising from just 1% in 2022 to 8% by 2024.

Despite our efforts to sample broadly, our analysis has several limitations. First, examining only 200 papers per year may yield estimates that deviate from the true prevalence of each trend across the entire NeurIPS corpus. Second, focusing exclusively on NeurIPS as the only conference introduces a potential bias, as NeurIPS submissions may not fully reflect developments in other leading venues or subfields. Third, we were only able to extensively validate the accuracy of results for small samples of studies during the iterative optimization of our prompt; for the overall experiments, we simply had to rely on the high performance of the optimized prompt and only skimmed each answer for its accuracy. For this, we evaluated the correctness of the provided one-sentence explanation with the content of the abstract of each paper, but did not (have to) change any answer given by the model. Finally, by excluding this year's conference submissions (NeurIPS 2025), whose acceptance outcomes still remain unknown, we are missing important continuing directions that either alter or, more likely,

Table 1: Numeric results of our quantitative analysis. Entries represent percentage of answer pairs to the following questions: 1. Does the study incorporate domain-specific expert knowledge about a problem into machine learning models to reduce the demand for computation and/or data? 2. Are large language models (LLMs) used for incorporating domain-specific expert knowledge as training or context data into problem solving?

| Response | NeurIPS 2020 | NeurIPS 2021 | NeurIPS 2022 | NeurIPS 2023 | NeurIPS 2024 |
|----------|--------------|--------------|--------------|--------------|--------------|
| No - No | 88 | 81.5 | 83 | 84.5 | 78 |
| Yes - No | 12 | 18.5 | 16 | 11 | 14 |
| No - Yes | 0 | 0 | 0 | 3.5 | 5.5 |
| Yes - Yes | 0 | 0 | 1 | 1 | 2.5 |

amplify our observed trends once it is clear which papers are accepted. Appendix A contains further detail about our conducted experiments.

## 6  Expanding on Sutton's Bitter Lesson under a decline in Moore's Law

The observations we present are not to be simply understood as a contradiction to Sutton's *Bitter Lesson*. In fact, they highlight the importance of general-purpose computation and the utilization of large amounts of data for advanced problem-solving with ML, as for example LLMs themselves are a product of these principles. However, one aspect of Sutton's *Bitter Lesson* we do want to contradict, is his perspective on expert domain knowledge being unimportant in light of growing computational resources and general-purpose *learning* and *search*. Instead, we highlight that the expanding scope of data, and in particular the emergence of powerful LLMs and multi-modal transformers, is increasingly turning human-like thinking and domain expertise themselves into valuable data for problem-solving.

Sutton leans his perspective on exponentially falling costs of computation due to Moore's Law. Today, however, we observe a clear slowdown and an approaching end to Moore's Law for digital computers, driven by insurmountable physical limitations, escalating costs, and diminishing performance returns from further miniaturizing silicon-based transistors. This challenge underscores the crucial need and increasing motivation for integrating expert domain knowledge in problem-solving with ML, in order to enhance computation and data efficiency in future solution methods.

A more subtle implication of Sutton's *Bitter Lesson* that we want to contradict, is the notion that simply increasing computational power and data consistently yields better results in ML-driven problem-solving. While this often holds under idealized conditions, such as assuming that data is *independent and identically distributed (i.i.d.)*, perfectly balanced, and noise-free, these assumptions are rarely met in real-world scenarios. Given the *law of large numbers*, larger datasets help us align sample distributions with true distributions, but this principle is far less applicable outside some of the well-structured domains like board games that Sutton discusses. In many real-world applications, such as the wide range of ML tasks related to tackling climate change [17], data is often non-i.i.d., noisy and highly imbalanced. In many of these applications, the presence of complex physical relationships between variables and spatio-temporal dependencies render simple assumptions about data and computation that is uninformed about these domain-specific relationships insufficient [18].

Other evolving areas of ML research like *active learning* also challenge the idea that more data alone leads to better outcomes, while supporting the importance of general-purpose computation [19]. Active learning focuses on the question of which data to collect for training a model that makes the best possible predictions when given a limited budget for data. For example, in the previously discussed study on electric load profile forecasting, an active learning approach achieves 26–81% higher accuracy while using 29–46% less data compared to traditional *passive learning* methods, without integrating any type of deep domain expertise about the problem into the ML model [11]. In a modern era of AI and ML, simply seeking more data and compute may increasingly become a naive approach.

Lastly, it is worth noting that our quantitative analysis reveals that only 12–18.5% of studies actively integrate problem-specific domain expertise beyond general-purpose learning and search. This is substantially less than one would expect in light of Sutton's Bitter Lesson and the attention that it has received by the AI and ML community.

# 7 Alternative views: consistencies and inconsistencies with other opinions

The interplay between human domain expertise and the scope of data in AI has long been contested, producing a rich tapestry of, and at times contradictory, perspectives [13, 14, 20–23]. As AI and ML research accelerates at an unprecedented rate, it becomes even more crucial to situate our perspective within this broader discourse, extending the conversation beyond Sutton's Bitter Lesson. In synthesizing these viewpoints, we see a landscape where data, expertise, and computational power are not opposing forces but complementary dimensions. Recognizing their interplay, and designing AI systems that integrate them thoughtfully, will be critical for advancing both the efficiency and the responsibility of future AI and ML research.

**Expert knowledge reduces data and compute requirements.** Our first observation emphasizes that integrating problem-specific expert knowledge into ML models can reduce the need for extensive computation and data. This view aligns with the principles of *informed ML*, which argue that prior knowledge, when judiciously integrated, enhances sample efficiency and directs learning toward more relevant abstractions [13]. It also echoes with Rodney Brooks' environmental critique of Sutton's stance, articulated in *A Better Lesson*, where he emphasizes that leveraging problem-specific structure can yield dramatic savings in data collection and training cost [22].

**Data's expanding boundaries via LLMs and multimodal models.** Our second key observation focuses on the historic expansion of the boundaries of what constitutes data for problem-solving in ML, and that the rise of powerful LLMs and multi-modal transformer architectures has notably contributed to this trend, allowing us to integrate increasing forms of expert domain knowledge and task contexts as data. This is supported by Dash et al. [14], who argue that domain knowledge can be represented not only through logical or numeric constraints but also through natural language statements and conversations, aligning with our view on the growing importance of LLMs for processing expert domain knowledge as data. However, our observation remains mostly different from that of Von Rueden et al. [13] who view 'prior knowledge' as something that accompanies data and who strictly separate its treatment from that of data.

**Chomsky's challenge and a human–AI synergy.** Another interesting counterpoint is Noam Chomsky's critique of concurrent LLMs, which he argues lack the capacity for human-like reasoning, explanation, and moral consideration [23]. Chomsky et al. [23] assert that LLMs fail to generate explanations and instead merely describe and predict, often incorrectly. While we share Chomsky's concerns about the limitations of ML systems more broadly, including LLMs, in addressing complex issues of morality and ethics, we diverge in our views that these systems, when combined with validated human domain expertise, can still significantly enhance our efficiency for solving critical problems with positive outcomes for humanity, such as tackling a wide range of climate change related ML tasks [17]. Consequently, we believe that concurrent LLMs merit the attention they are receiving for their potential to complement human reasoning and drive innovation.

**Contextualizing Data and "Situated Knowledge".** Our stance aligns with Boyd and Crawford's viewpoint about the importance of context in which data is collected and analyzed, and that the reduction of data to numeric values for fitting into a rather narrow model can strip it of the nuances that are critical for meaningful analysis [21]. It further resonates with Haraway's notion of 'situated knowledge', which presents the idea that knowledge is always partial and shaped by the context and position of the 'knower'. Haraway emphasizes that expertise is not just a static body of knowledge but is constructed and meaningful within specific contexts, thereby supporting the importance of incorporating context information as data for more sophisticated ML solutions [20].

# 8 Conclusion

In the rapidly evolving landscape of AI and ML, we advocate that it is essential to recognize human expertise as data. This is information that often resides in the form of mathematical equations, computer code, or natural language descriptions of domain insights that are critical for solving tasks. Such information is typically not recognized as first-class data by our current conventions.

We emphasize that incorporating human expertise into problem-solving in ML often provides valuable structure to solution methods and is integral to reducing our demand for computation and data, as well as enhancing out-of-distribution generalization. Additionally, we emphasize that the scope of data in ML is expanding over time and typically enhances the inductive power of ML models. In this context,

we highlight the emergence of LLMs as the latest leap forward in this trend, which increasingly allows us to use human expertise as data for general-purpose computation and problem-solving.

In order to fully realize the potential of AI and ML models in light of these advancements, we call upon the AI and ML community to rethink our existing approaches to data collection and model architecture design. Much of our efforts in improving the collection and documentation of data with standards like the FAIR principles [24] or the newer Croissant format [25] focus on enhancing the standardized structure and comprehensiveness of metadata, with the primary goal of making datasets more interoperable and discoverable by tools and platforms. This is attributed to the assumption that *"...data documentation written in natural language, without a standard machine-readable representation ...makes data documentation challenging for machines to read and process"* [25], a clearly outdated view in light of the capability of concurrent LLMs.

In addition to suggested data documentation standards, we encourage practitioners to include detailed contextual information and task-specific insights as an integral part of their datasets. This information should be accessible via the same Application Programming Interface (API) that provides dataset features and labels. We propose that such contextual information could encompass some of the metadata fields introduced in recent standards, such as those in the "semantic" and "structural" layers of Croissant [25]. Rather than adhering to a rigid format, this context can be effectively communicated in natural language, compiled into a single text file referred to as a "task description", and thereby provide valuable flexibility for data documentation alongside well-structured metadata standards.

The advancement of LLMs and multimodal architectures in using human expertise as general-purpose computational data simultaneously creates a strong dependence on the availability of validated, well-curated datasets representing domain-specific expertise. Expanding the number of these curated repositories, containing validated expert knowledge and targeted insights, can significantly reduce the cost and dependency on human input, particularly during model training and fine-tuning with RLHF. The increasing availability of these datasets can shift more computational burden towards supervised and unsupervised learning components of the training pipeline, and decrease the relative burden on human feedback within the overall training workflow of advanced models.

We envision these changes to give rise to powerful domain-expert ML models, as well as mixtures of these, and to deepen the integration of human expertise into automated problem-solving. By embedding LLMs within streamlined, multi-modal architectures that consistently combine language with task-specific numeric data, we envision that a significant number of conventional ML solution methods will be enhanced.

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

Table 2: Detailed results of our quantitative analysis by response and subcategory across NeurIPS 2020–2024. Entries represent either percentage or extrapolated count of answers to the following pair of questions: 1. Does the study incorporate domain-specific expert knowledge about a problem into machine learning models to reduce the demand for computation and/or data? 2. Are large language models (LLMs) used for incorporating domain-specific expert knowledge as training or context data into problem solving?

| Response | 2020 | 2021 | 2022 | 2023 | 2024 |
|---|---|---|---|---|---|
| **Papers 1–500** | | | | | |
| No – No | 88 | 80 | 85 | 84 | 78 |
| Yes – No | 12 | 20 | 15 | 10 | 11 |
| No – Yes | 0 | 0 | 0 | 4 | 7 |
| Yes – Yes | 0 | 0 | 0 | 2 | 4 |
| **Papers 501–1000** | | | | | |
| No – No | 88 | 83 | 81 | 85 | 78 |
| Yes – No | 12 | 17 | 17 | 12 | 17 |
| No – Yes | 0 | 0 | 0 | 3 | 4 |
| Yes – Yes | 0 | 0 | 2 | 0 | 1 |
| **Percentage** | | | | | |
| No – No | 88 | 81.5 | 83 | 84.5 | 78 |
| Yes – No | 12 | 18.5 | 16 | 11 | 14 |
| No – Yes | 0 | 0 | 0 | 3.5 | 5.5 |
| Yes – Yes | 0 | 0 | 1 | 1 | 2.5 |
| **Extrapolated** | | | | | |
| No – No | 1687.84 | 1902.21 | 2411.15 | 3028.48 | 3539.64 |
| Yes – No | 230.16 | 431.79 | 464.80 | 394.24 | 635.32 |
| No – Yes | 0.00 | 0.00 | 0.00 | 107.52 | 249.59 |
| Yes – Yes | 0.00 | 0.00 | 29.05 | 35.84 | 113.45 |

## A  Additional details about conducted experiments

We have conducted our experiments in two batches, each containing 100 studies per year. Table 2 shows the results for these splits and their overall percentages. The table further contains results that were extrapolated by the total number of papers published in each conference. These numbers are deduced by multiplying percentages with 19.18 for NeurIPS 2020, 23.34 for NeurIPS 2021, 29.05 for NeurIPS 2022, 35.84 for NeurIPS 2023, and 45.38 for NeurIPS 2024. Figure 1 further shows a visualization of percentages and extrapolated results. The supplementary material further contains a documentation of responses for each study, organized in the same batches as seen in Table 2.

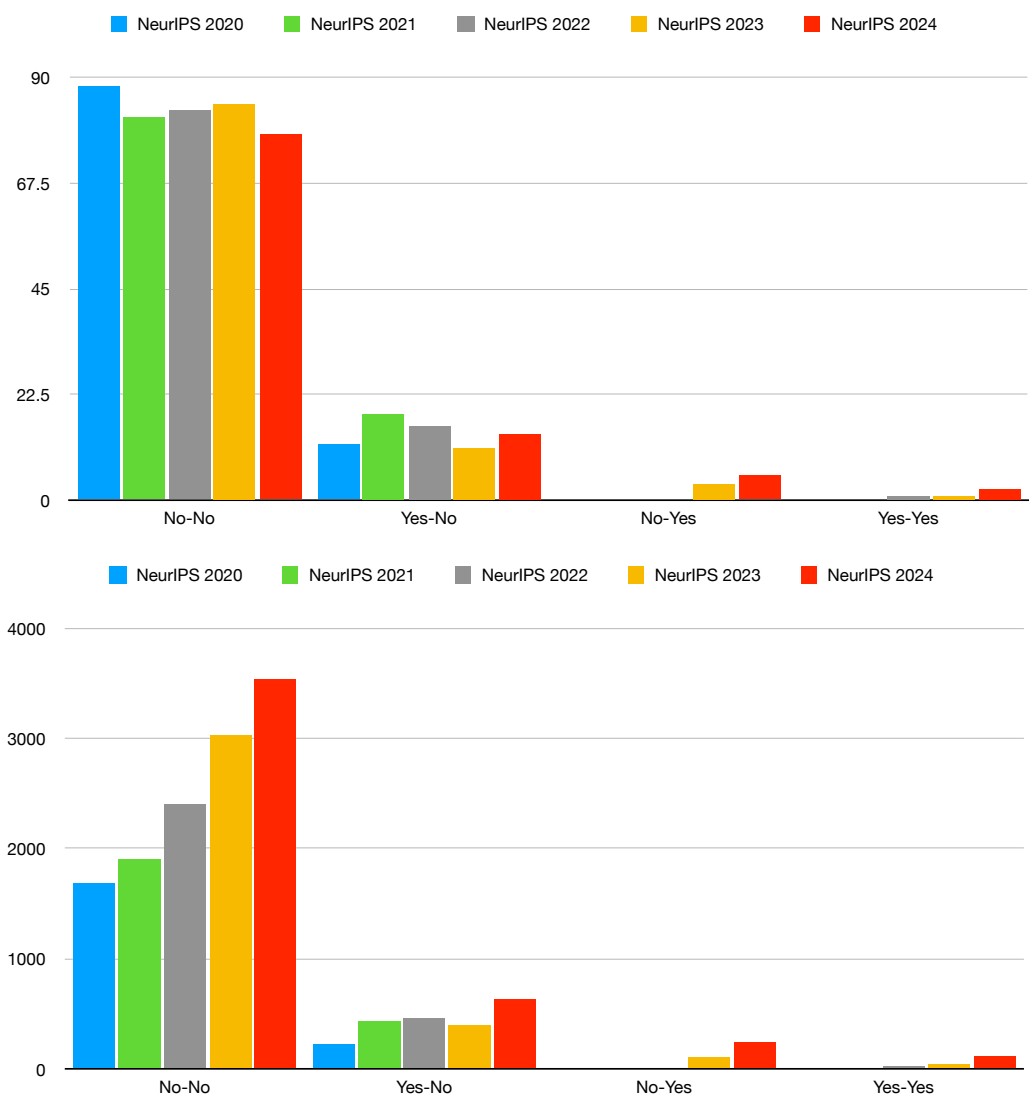

Figure 1: Results of our quantitative analysis of NeurIPS 2020-2024 papers. Upper figure bars show percentage and lower figure bars extrapolated count of total studies, of answers to the following questions: 1. Does the study incorporate domain-specific expert knowledge about a problem into machine learning models to reduce the demand for computation and/or data? 2. Are large language models (LLMs) used for incorporating domain-specific expert knowledge as training or context data into problem-solving?

