# OpenReview forum: "From Bitter to Better Lessons in AI: Embracing Human Expertise as Data"
_NeurIPS.cc/2025/Position_Paper_Track — Submitted to NeurIPS 2025 Position Paper Track_

### Official Review · Reviewer_5pY9 · 2025-07-23

**Significance:** 3
**Presentation:** 2
**Rating:** 4
**Confidence:** 3

**Summary:**

This paper challenges Sutton's "Bitter Lesson" by arguing that human expertise should be treated as "first-class data" in AI systems. The authors claim domain knowledge integration reduces computational requirements through physics-informed ML, geometric deep learning, and safe RL examples. They position LLMs as enabling expertise integration via RLHF and in-context learning. A survey of 1,000 NeurIPS papers shows low explicit domain integration (12-18%) but rising LLM adoption (1% to 8%). The paper concludes by advocating new data documentation standards.

**Strengths:**

1. Addresses a timely and relevant debate about scaling versus domain knowledge in current AI development
2. Provides comprehensive coverage across multiple ML subfields with concrete examples like nanophotonic device optimization
3. Offers quantitative analysis of NeurIPS trends, providing useful empirical data despite limitations

**Weaknesses:**

1. The terms “human expertise,” “domain knowledge,” and “contextual information,” are used interchangeably without clear definitions. This lack of precision weakens the argument, making it difficult to define what the authors mean by "expertise as data."
2. The NeurIPS analysis appears to rely on ChatGPT for paper classification, with limited detail on manual validation or inter-rater checks. Statistical measures like confidence intervals or significance testing are also not reported.
3. The paper highlights successful cases of domain knowledge integration, but does not discuss instances where such approaches may have been less effective or impractical. Without examples of failure or limitations, assessing the broader applicability or opportunity costs of integrating expert knowledge becomes challenging.
4. In Section 2, the authors argue that integrating domain knowledge “narrows the hypothesis space” and improves efficiency. However, they do not discuss potential downsides of this narrowing — such as excluding hypotheses that contradict current expert understanding. This is especially relevant in fields where innovation has historically come from outside dominant frameworks.

**Questions:**

1. Who determines what constitutes valid "expertise" and how do we validate expert claims when experts disagree?
2. How do we prevent historical biases in expert knowledge from becoming architectural features?
3. What happens to AI democratization if model building requires deep domain expertise?

**Alternative Position:**

Yes, and alternative positions are well-considered and addressed by the argument

**Author Identification:**

No.

**Context:**

2

**Discussion:**

3

**Ethics:**

["NO or VERY MINOR ethics concerns only"]

**Position:**

Yes, the paper argues for or against a position related to machine learning.

**Support:**

2

**Thoroughness:**

4

---

### Official Review · Reviewer_4izG · 2025-08-01

**Significance:** 4
**Presentation:** 4
**Rating:** 7
**Confidence:** 4

**Summary:**

This paper argues that human expertise should be regarded as an important form of data in machine learning. Integrating such expertise can reduce the hypothesis space, decrease the amount of training data required, and improve out-of-distribution generalization capabilities. The authors demonstrate this point by reviewing multiple subfield cases, showing that incorporating domain insights can enhance efficiency and performance. LLMs are seen as a typical example of leveraging human knowledge as data—through human feedback and prompts, LLMs utilize information provided by humans to solve tasks. The authors also analyzed NeurIPS papers from 2020 to 2024 and found that approximately 15% of papers explicitly integrated domain knowledge, although this proportion is gradually increasing through LLM-based methods. They contrast this with Rich Sutton's “bitter lesson” and point out that with the slowdown in hardware progress, now is the time to leverage human insights as data. Finally, the paper outlines specific steps to achieve this vision, such as improving dataset documentation and establishing curated knowledge repositories.

**Strengths:**

1. The paper's central argument (treating human expertise as data) is clearly articulated and consistently maintained throughout.
2. Timely topic: The paper addresses pressing challenges in the field of machine learning (e.g., how to improve generalization capabilities as data scales reach bottlenecks) and connects its arguments to current trends (such as the rise of large language models (LLMs)), making the discussion highly relevant.
3. This position is supported by evidence from multiple subfields and a quantitative analysis of 1,000 NeurIPS papers, adding credibility to the argument.
4. The authors acknowledge and address alternative viewpoints (particularly Sutton's “bitter lessons”), strengthening their argument by thoroughly considering counterarguments.
5. The paper goes beyond theoretical considerations to propose concrete recommendations (e.g., improving dataset documentation to include context, establishing curated knowledge bases, and designing models that incorporate expert opinions), making its ideas practically actionable.

**Weaknesses:**

1. The study primarily consists of opinions and reviews, lacking original experiments to directly prove its arguments. It relies on literature cases and reasoning, which may not satisfy readers expecting specific new evidence.
2. The paper provides little detail on how to systematically collect and integrate expert insights, or how to mitigate biases in human knowledge. This practical gap leaves unclear whether this vision is feasible in the real world.
3. The analysis of 1,000 NeurIPS papers is inadequately explained; the reliability of the reported statistics is questionable due to the lack of a clearly defined standard for “domain expert integration.”

**Questions:**

1. How did you analyze 1,000 NeurIPS papers? What criteria did you use to determine whether a paper “explicitly” incorporated domain expert knowledge, and how did you ensure that these criteria were applied consistently?
2. Human expert knowledge may sometimes be biased, incomplete, or erroneous. How do you suggest ensuring the quality and fairness of expert knowledge when using it as data? For example, have you considered verification steps or guidelines to prevent flawed or biased expert information from influencing the model?
3. Given that large models trained on internet text have already implicitly absorbed a significant amount of human knowledge, what are the primary advantages of explicitly integrating curated expert knowledge? In which specific scenarios would a model enhanced with explicit expert input significantly outperform one relying solely on broad, uncurated data?

**Alternative Position:**

Yes, and alternative positions are well-considered and named but not addressed

**Author Identification:**

No.

**Context:**

4

**Discussion:**

3

**Ethics:**

["NO or VERY MINOR ethics concerns only"]

**Position:**

Yes, the paper argues for or against a position related to machine learning.

**Support:**

3

**Thoroughness:**

3

---

### Official Review · Reviewer_cfp5 · 2025-08-12

**Significance:** 2
**Presentation:** 3
**Rating:** 5
**Confidence:** 3

**Summary:**

This paper challenges the traditional view in artificial intelligence (AI) and machine learning (ML) that treats data solely as clean, numerical features and labels. It argues that human expertise, encoded in natural language, mathematics, and software, should also be considered a vital form of data. The authors demonstrate that embedding expert knowledge into ML can reduce computational complexity, improve generalization, and make models more efficient. They explore how large language models (LLMs) can integrate human domain expertise, presenting examples like reinforcement learning with human feedback and in-context learning. The paper also examines recent trends in AI, highlighting the increasing role of LLMs in solving problems. The authors advocate for a shift in AI practices to incorporate human expertise as data, improving AI's problem-solving efficiency and expanding its potential.

**Strengths:**

- effectively presents a compelling argument for integrating human expertise as data in AI and ML, challenging the prevailing view that data should only be numerical.

- Clearly articulate how incorporating expert knowledge can reduce computational demands and improve generalization in ML models. They support their position with relevant examples from areas like physics-informed ML, geometric deep learning, and reinforcement learning with human feedback, showing how these approaches enhance efficiency and effectiveness.

- The use of large language models (LLMs) as a method to incorporate human expertise is well-documented and demonstrates their increasing importance in AI research.

**Weaknesses:**

- While the paper provides a strong case for integrating human expertise as data, it could benefit from a deeper exploration of potential challenges or limitations in this approach. For instance, the feasibility of consistently encoding human expertise across diverse domains may require significant efforts in standardization and validation, which the paper does not fully address.

- Alternative positions, such as emphasizing hybrid models that combine computational power with selective expert input or exploring the potential of unsupervised learning and active learning to reduce reliance on human expertise, are not fully explored.

- Furthermore, the paper could engage more with critiques from those who argue that increased computational power and data-driven approaches, even without human expertise, will eventually outperform expert-driven methods in many domains.

**Questions:**

see Weaknesses

**Alternative Position:**

No

**Author Identification:**

No.

**Context:**

3

**Discussion:**

3

**Ethics:**

["NO or VERY MINOR ethics concerns only"]

**Position:**

Yes, the paper argues for or against a position related to machine learning.

**Support:**

3

**Thoroughness:**

3

---

### Note · Authors · 2025-08-26

**1-11 Submit Again:**

Probably yes

**1-1 Submission Process:**

5

**1-2 Next Year:**

Same concept. It seems like a good one to us.

**1-3 Future Development:**

A more aligned deadline with the remaining conference and more specific guidelines for authors. Currently, author guidelines have been a bit scattered.

**1-4 Interest:**

["Panel discussions with other position paper authors", "Structured debates on controversial topics", "Mentorship programs for early-career researchers"]

**1-4 Other Interest:**

I would be interested in participating as a reviewer or area chair.

**1-5 Thoughtful:**

7

**1-6 Supportive:**

7

**1-7 Technical Aspects Versus Position:**

7

**1-8 Gate Keeping:**

10

**1-9 Camera Ready Changes:**

We thank all reviewers for their thoughtful feedback. Their comments have helped us identify important changes that will improve the rigor of our paper. If accepted, our camera-ready version will include the following changes:

Change 1: Expanding discussion of expertise integration with Retrieval-Augmented Generation.
Currently, we describe two core mechanisms by which LLMs integrate human expertise as data: (1) through in-context learning and (2) through training data with RLHF. We propose to add a third, equally important approach: Retrieval-Augmented Generation (RAG).

Change 2: Defining human expertise and its integration pathways.
We will add a new section after the introduction that explicitly defines human expertise and categorizes its different forms (e.g., mathematical formalism, algorithms, domain-specific heuristics, natural language explanations). In this section, we will also clarify how human expertise is operationalized in ML systems, for example through loss/reward functions, architecture design, data preprocessing choices, and more. This section will strengthen conceptual clarity and address reviewer concerns about terminology precision.

Change 3: Conducting additional experiments and strengthening experimental rigor. We would like to run additional experiments leveraging a more detailed taxonomy of human expertise as introduced after Change 2, and measure the distribution of NeurIPS papers using these. We further plan to introduce inter-rater checks both across multiple LLMs, and among human evaluators to verify classifications. Where appropriate, we will incorporate confidence intervals and significance testing to bolster empirical reliability.

Change 4: Discussion of limitations and risks of incorporating human expertise as data into ML
We plan to add a section on the limitations, risks and potential downsides that the treatment of human expertise as data entails. The missing discussion of this is a concern raised by all three reviewers.

**3-1 Review Response1:**

cfp5

**3-2 Reaction To Review1:**

Issue 1
“This paper challenges the traditional view... embedding expert knowledge can reduce computational complexity...”
Correct.

Issue 2
“They explore how large language models (LLMs) can integrate human domain expertise...”
Correct. We will also add Retrieval Augmented Generation (RAG) as a third essential method for integrating expertise.

Issue 3
“The paper also examines recent trends... advocating a shift in AI practices...”
Correct.

Issue 4 (Strengths)
“Compelling argument... clear articulation... supported with relevant examples...”
We thank the reviewer for recognizing these strengths.

Issue 5 (Challenges of encoding expertise)
“The paper could benefit from a deeper exploration of potential challenges...”
Response: We agree. We will add a section before the conclusion explicitly addressing challenges such as standardization, validation, transferability, and risks (e.g., bias, overfitting, constrained innovation).

Issue 5 (Alternative positions)
“Hybrid models, unsupervised and active learning not fully explored.”
We agree these are important complements. We will more explicitly emphasize how active and unsupervised learning reduce reliance on expert input by leveraging general-purpose computation, noting existing discussion in Sections 3 and 7 and expanding with clearer articulation of their role in hybrid approaches.

Issue 6 (Critique: scale over expertise)
“The paper could engage more with critiques... that scale alone will outperform expert-driven methods.”
We agree. While scaling has been successful, we argue expertise reduces sample/compute needs, improves interpretability, and mitigates risks like hallucination. In revision, we will expand our taxonomy of human expertise, highlighting how domain knowledge is already implicitly embedded (e.g., data preprocessing choices, reward/loss design). This will help us directly engage with critiques favoring scale-only approaches.

**3-3 Review Response2:**

4izG

**3-4 Reaction To Review2:**

Issue 1
“Paper argues that human expertise should be treated as data...”
Correct.

Issue 2 (Strengths)
“Central argument is clear, timely, supported by evidence, considers counterarguments, and offers concrete recommendations.”
We thank the reviewer for recognizing these strengths.

Issue 3
“Relies on reasoning and literature, lacks original experiments.”
We acknowledge this. As a position paper, our aim is conceptual contribution, supported by trends and literature. Our analysis of 1,000 NeurIPS papers is the main empirical element; full experiments are beyond scope.

Issue 4
“Unclear how to systematically collect expertise or mitigate biases.”
Agreed. We will add a section on limitations, covering challenges of standardization, disagreement, historical bias, and practical strategies for validation.

Issue 5
“NeurIPS analysis inadequately explained; unclear standard for ‘domain expert integration.’”
Agreed. We will expand our taxonomy of expertise and define integration criteria more clearly to improve reliability of statistics.

Issue 6
“How did you analyze 1,000 papers?”
We sampled 200 papers per year (2020–24) and processed them with an LLM prompt, followed by manual checks. We note limitations of full validation and will strengthen consistency with our expanded taxonomy and inter-rater checks across LLMs and humans.

Issue 7
“Expert knowledge may be biased or incomplete. How ensure fairness?”
We agree. We will emphasize plural representation, inter-rater checks, transparent documentation, and validation steps to reduce bias and error.

Issue 8
“Large models already contain human knowledge. Why explicit expert integration?”
While internet-scale training captures broad knowledge, it is noisy and inconsistent. Curated expertise provides higher-quality signals, essential for high-stakes fields (e.g., medicine, law, engineering), dynamic regulatory domains, and specialized sciences. RAG systems already show advantages of curated repositories over uncurated text.

**3-5 Review Response3:**

5pY9

**3-6 Reaction To Review3:**

Issue 1
“Challenges Sutton’s ‘Bitter Lesson’ by treating expertise as data.”
Correct. We expand Sutton’s view, clarifying in Sec. 6 that expertise complements scaling rather than replaces it.

Issue 2
“Examples show reduced compute via physics-informed ML, etc.”
Correct, but not universal. We will note these are illustrative cases, not general rules.

Issue 3
“LLMs integrate expertise via RLHF and in-context learning.”
Correct. We will add Retrieval Augmented Generation (RAG) as a third mechanism.

Issue 4
“Survey shows low explicit domain use, rising LLM adoption, concluding with standards advocacy.”
Correct. We will clarify findings and implications with revised experiments.

Issue 5 (Strengths)
We thank the reviewer for recognizing the timeliness, coverage, and empirical contribution.

Issue 6
“Terms used interchangeably, weakening argument.”
Agreed. We will standardize on “human expertise” and introduce a definition section.

Issue 7
“Classification relied on ChatGPT; limited validation and no stats.”
Agreed. We will add human/LLM inter-rater checks, expand validation, and report confidence intervals.

Issue 8
“No discussion of failures or risks; narrowing may exclude discoveries.”
Agreed. We will add a section on risks, including overconstraint and missed innovations, with concrete examples.

Issue 9
“Who defines valid expertise when experts disagree?”
Consensus-based areas (e.g., physics) are reliable, but where contested we advocate plural representation, transparent documentation, and inter-rater checks.

Issue 10
“Risk of embedding historical bias in architectures.”
Agreed. We will discuss this in the new limitations section, highlighting corrigible/updated repositories and plural perspectives as mitigation.

Issue 11
“Risk to democratization if expertise is required.”
We argue the opposite: structured expertise lowers barriers by enabling smaller labs to leverage curated knowledge, modular data, and community-driven curation.

---

### Meta-Review · Area_Chair_hU5h · 2025-09-11

**Rating:** 5
**Confidence:** 1

**Strengths:**

Clear position presentation: disagreeing with Sutton's "the bitter lesson" and arguing for human experience as the "new data".

A meta-analysis on NeurIPS 2020 - 2024 papers on whether human and domain-specific insights were incorporated into technical designs of the machine learning methodology (and whether LLMs were used for this purpose). This analysis is intended to support the argument that LLMs can help the incorporation of human experiences and/or domain knowledge.

**Weaknesses:**

Rich Sutton has updated his belief (together with David Silver) in a new opinion piece "Welcome to the Era of Experience" https://storage.googleapis.com/deepmind-media/Era-of-Experience%20/The%20Era%20of%20Experience%20Paper.pdf, which argued similar points but with more extended discussions on RL related aspects. This essay has already generated discussions within the field of RL and Generative AI. Given the impact of this recent essay that has overlapping argument with the submitted paper, I feel it would be important for the authors to discuss Sutton's latest opinion and perhaps signpost explicitly the unique aspects of this paper's arguments.

The meta-analysis is conducted by prompting ChatGPT with instructions. While this approach is understandable given the workload of analysing ~1000 selected papers at NeurIPS 2020 - 2024, the correctness of the ChatGPT generated analysis results is questionable. The authors stated that they manually validated the correctness of all responses, but they did not present precisely how they performed the validation as well as the final configuration's FPR and FNR rates.

**Questions:**

See "weakness" part.

**Thoroughness:**

3

---

### Decision · Program_Chairs · 2025-09-26

Reject